# Smart Agent System for Cyber Nano-Manufacturing in Industry 4.0

**Naif Almakayeel** [1,*] **, Salil Desai** [2] **, Saleh Alghamdi** [1] **and Mohamed Rafik Noor Mohamed Qureshi** [1]

1 Industrial Engineering Department, King Khalid University, Abha 62529, Saudi Arabia; syalghamdi@kku.edu.sa (S.A.); mrnoor@kku.edu.sa (M.R.N.M.Q.)
2 Department of Industrial and Systems Engineering, North Carolina A&T State University, Greensboro, NC 27411, USA; sdesai@ncat.edu
* Correspondence: halmakaeel@kku.edu.sa

**Abstract:** The development of Cyber-Physical Systems (CPS) and the Internet of Things (IoT) has influenced Cyber-Physical Manufacturing Systems (CPMS). Collaborative manufacturing among organizations with geographically distributed operations using Nanomanufacturing (NM) requires integrated networking for enhanced productivity. The present research provides a unique cyber nanomanufacturing framework by combining digital design with various artificial neural networks (ANN) approaches to predict the optimal nano/micro-manufacturing process. It enables the visualization tool for real-time allocation of nano/micro-manufacturing resources to simulate machine availability for five types of NM processes in real-time for a dynamic machine identification system. This research establishes a foundation for a smart agent system with predictive capabilities for cyber nanomanufacturing in real-time.

**Keywords:** artificial neural network (ANN); cyber-physical systems (CPS); Industry 4.0; Internet of Things (IoT); nanomanufacturing; smart expert system

## 1. Introduction

In recent years, the need to automate nanomanufacturing processes has increased with the demand for more choices in commercial products. There is also a growing diversity of products that manufacturers seek to transform into different types of nanoscales. Quality control is an important challenge faced by nanomanufacturing experts. Implementations of the Internet of Things (IoT), such as cyber-physical systems (CPSs), have positively affected nanoscale processing [1]. To obtain satisfactory nanoscale products, the proper processes or techniques need to be accurately chosen. Precise analysis plays a crucial role in production lines, at least as important as material considerations. When it comes to selecting facilities for the optimum workplace, nanomanufacturing costs must be factored in, as they need a perfect configuration or system. It needs a perfect place to have adequate production. In other words, handling dynamics, time, and simultaneousness in heterogeneous (interconnected) arrangements is a primary engineering issue. However, the quantity and difficulty of intelligence are developing so quickly that programming executions are the central part of system design, effectiveness, and eventual confirmation. CPSs are essential for the future of the system industry worldwide, and the ability to interface at all levels is required from application engineers to outcome designers, from device manufacturers to technology suppliers, and from service to research.

CPSs have been contributing greatly to leading advancements in critical areas of systems control and research [2]. The integration of computer-embedded systems with physical processes calls for higher precision levels. CPSs have made it possible for notable advancements to be made in areas such as distributed robotics, systems of defense, and electric power infrastructure control that need a high level of security, reliability, and accuracy owing to the unpredictable nature of the physical world [3].

Nanoscale manufacturing has seen recent advancements in the design of control processes and systems [4,5]. Consequently, the cyber systems in use within these scales have been the recent focus for designers [6]. For instance, a nano-robot with sizes ranging within the nanoscale would be preferred as a system that would achieve most of the programmed functions at the nanoscale [7]. Such systems would have the desired advantages of flexibility and access to very small areas [8]. Moreover, the systems would be less costly, more distributed, adaptive, and more robust [9]. The current motivation of designers is to improve on this miniaturization even further to enhance the integration capabilities for communication, sources of power and strategies, and tools of computation [10]. So, it becomes an opportunity and a key priority for designers to overcome the associated challenges and prototype such system requirements in real life during design scenarios since the process is typically a series of trials with varied conditions until the desired result can be achieved [3]. Before the actual design, it is judicious to use system models to represent the required scenario. Thus, it is important to focus on the general concept of manufacturing cyber-products with the use of a nano-manufacturing approach [11].

In a cyber-nano manufacturing system, several issues may arise that are different from those of traditional cyber manufacturing [7]. Some of these issues include the need for multiple-scale design, simulation, and modeling of nano-systems. This entails the development of the techniques and tools necessary for making scalable processes used in nano-manufacturing. Cyber nano-manufacturing must meet a set validation, certification, and verification procedures since the level of expected precision is higher. These issues have been raised and have been the key focus of modern nanoscale systems [12]. Compared to traditional cyber manufacturing, there is more adaptability, complexity, and multi-functionality expected of the nano-manufacturing process. The response of the cyber nano-manufacturing systems must have an elastic response to external stimulus, better intelligence, greater autonomy, and smartness than the traditional systems [13]. These expectations of the cyber process of nano-manufacturing can be achieved by identifying a suitable solution for the realization of CPSs that are miniaturized with all physical components integrated and automated using computer-aided approaches. This way, the computation ability would be boosted, as well as the integration capacity. So, high yields of output would be realized [14].

The main challenge for achieving nanometer-scale products will be the ability to interconnect and interface with the macroscale world [15]. Thus, a fully scalable nanomanufacturing platform is required to ease the multiscale integration from the nanoscale to the microscale in addition to the macroscale [16]. It is necessary for these nanoscale interconnects to be fast, reliable, cost-effective, consume little power, and be able to link structures of different types, materials, and sizes [17]. Metrology is essential to guarantee that the nano-devices are suitable before and after connection [18].

Different manufacturing segments are going cyber because the rapid expansion of internet connections to machines has provided accessibility to a lot of different equipment and manufacturing technologies [19]. In the past, finding solutions might be limited to the expertise of a few people or specific industries. However, technological advancements have now made it possible to have a nanomanufacturing service hub where everybody can access equipment connected to the specialties [20]. Nevertheless, this abundance of available information and systems behooves users to identify the most suitable techniques for specific jobs based on input requirements such as throughput, accuracy, etc. The proliferation of internet connections includes equipment at the nanoscale, creating great potential for cyber manufacturing. Each of the nanomanufacturing techniques offers valuable technology, specialization, and expertise, but their availability varies widely. On reviewing the literature on the evaluation of machine availability using IoT, API, and Node-Red, little research is available in the area of machine availability in real-time. Hence, there is a need to design and develop a system for accurately identifying and efficiently meeting machine availability requirements.



For this reason, we are developing a system designed to efficiently put everything together [21]. Thus, when an input requirement comes, the system analysis using Artificial Intelligence (AI) can readily adapt to the needs. Then, the results are sent to the next part of the framework. Thus, we have built an entire framework that has a cyber nanomanufacturing architecture. It can draw from the knowledge base that has been generated based on literature reviews, experts' opinions, and original equipment manufacturers' (OEMs) data [22]. We also included CAD designs and the information coming from both the Node-Red and the IoT center connected to this equipment, resulting in a combination of all this information. So, based on the input provided and what equipment should be used, the equipment has the problem. Thus, we must determine (state?) the resource requirement and Node-Red brings this information, and all of these are interfaced to produce the needed output [23].

The feasibility of gathering and handling data is becoming easier with the advancement of Message Queuing Telemetry Transport (MQTT) and Node-Red. Furthermore, the ongoing advancement and integration of the IoT have led to the Industrial IoT (IIoT) for industrial advantages [24]. Badii et al. [24] developed a framework for controlling and supervising operations using MicroService Industry 4.0 scenarios. Various internet-based technologies related to cyber-physical systems, the IoT, cloud computing, Industrial Integration, and Industrial Information Integration are employed in Industry 4.0 [25]. Ferreira et al. [26] provided the framework for linking simulation and design with simulation classification for Industry 4.0. Cadavid et al. [27] modeled the complexity of using IoT in collecting data to adapt to manufacturing system changes. Dolgui et al. [28] surveyed customized assembly systems for Industry 4.0. There is a bright future for various nano-machining techniques in the electronic industries, biomedical industries, etc. [29]. An open-source IoT and blockchain-based peer-to-peer energy trading platform using ESP32-S2, Node-Red, and MQTT protocol was developed [30,31]. Taking account of the advancement in nano/micro-machining, the machine identification in real-time for various nanomachining processes is yet to be developed to help various industries. This paper describes the development of a cyber nanomanufacturing framework to integrate the design and manufacturing of nanoscale components and devices over cyberspace. The framework consists of three sub-systems which include: (1) Artificial Neural Network (ANN)-based Expert System, (2) a Cyber Interface Simulator, and (3) a Dynamic Nano-M/C Identification System.

## 2. Methodology

In the present research, a computer-integrated, manufacturing-system-based integration approach has been employed, where various internet-based techniques such as CPS, IoT simulator, Node-Red, and API have been integrated to find the machine availability in real-time. The system integration in the present research is depicted in Figure 1.

Cloud manufacturing enables the sharing of manufacturing resources with a wider customer base [32]. The implementation of CPSs in nanomanufacturing has been limited due to the complexity of high-end nano- and micro-manufacturing processes. This is based on the fact that each input part design with nano/micro-components has variations in topological features, process throughput, and resolution requirements. Moreover, the part design and user requirements need to be complemented with the process capability data of the machines. Thus, the selection of the optimal nano/micro-manufacturing process for each input part design is a complex, multicriteria decision-making problem with no readily applicable system available for cloud-based NM [33].

Figure 2 shows a schematic of the cyber nanomanufacturing framework. It consists of input part designs, a knowledge base, an IoT device interface, a smart cyber agent, and a dynamic machine identification system [34]. The input part characteristics were extracted from the nano-designs and user specifications. A knowledge base was populated based on NM literature review [35–37], subject matter feedback [38,39], and best practices from NM OEMs [40–42]. The CAD designs, user requirements, and knowledge base inputs were

preprocessed into a datasheet to be fed to the smart cyber agent. Three different ANN algorithms, which include the general regression neural network (GRNN), probabilistic neural network (PNN), and the backpropagation neural network (BPNN), were implemented to classify each part design with their respective nano/micro-manufacturing processes. An IoT simulator was modeled using Node-Red for the acquisition of machine availability and process capability data in cyberspace [43]. An application program interface (API) was developed to dynamically allocate resources over the cyber network by integrating inputs from the ANN model and the IoT simulator.

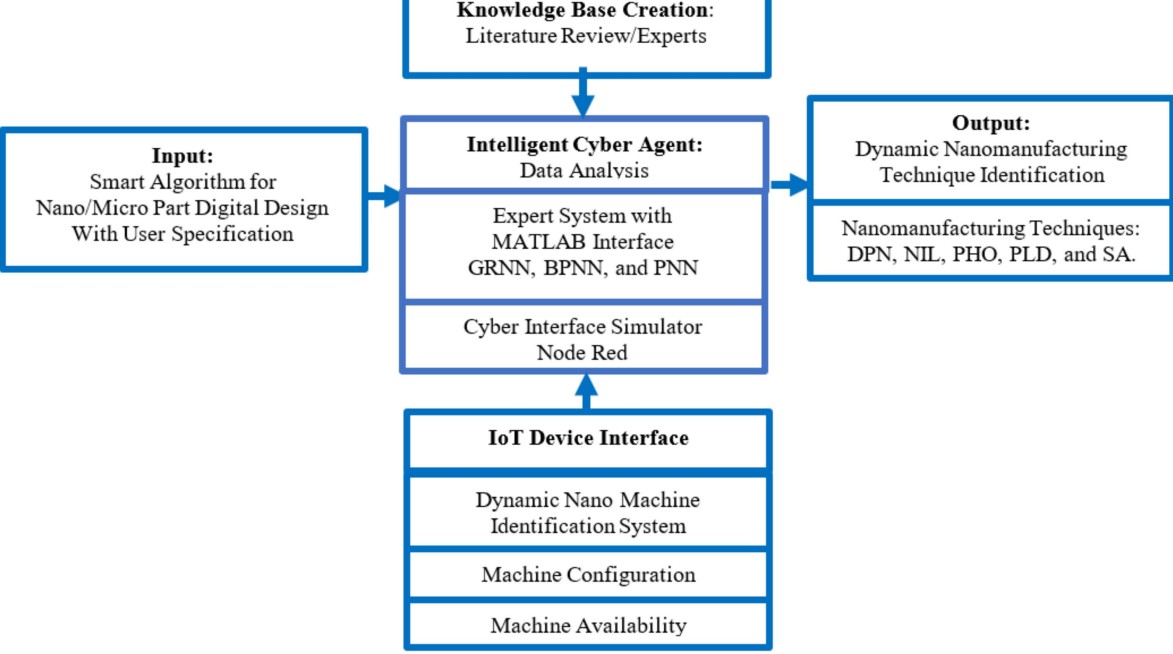

**Figure 1.** Research Methodology (Authors' work).

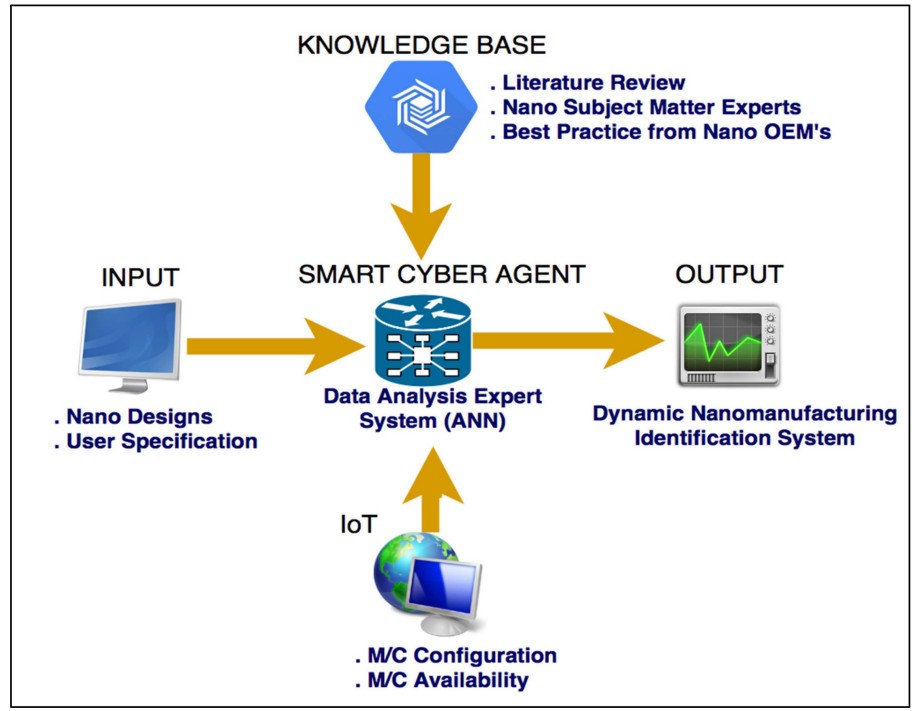

**Figure 2.** Framework for cyber nanomanufacturing (Authors' work).

### 2.1. ANN-Based Expert System

Figure 3 shows the procedure for extracting requisite information from the CAD nano designs, user inputs, and knowledge base to populate data sets for ANN. The data sets are further divided into training and test sets to be evaluated by three different ANNs. After executing the algorithms for a certain number of epochs the results were analyzed to fine-tune the ANN models for higher fidelity predictions of optimal nanomanufacturing processes.

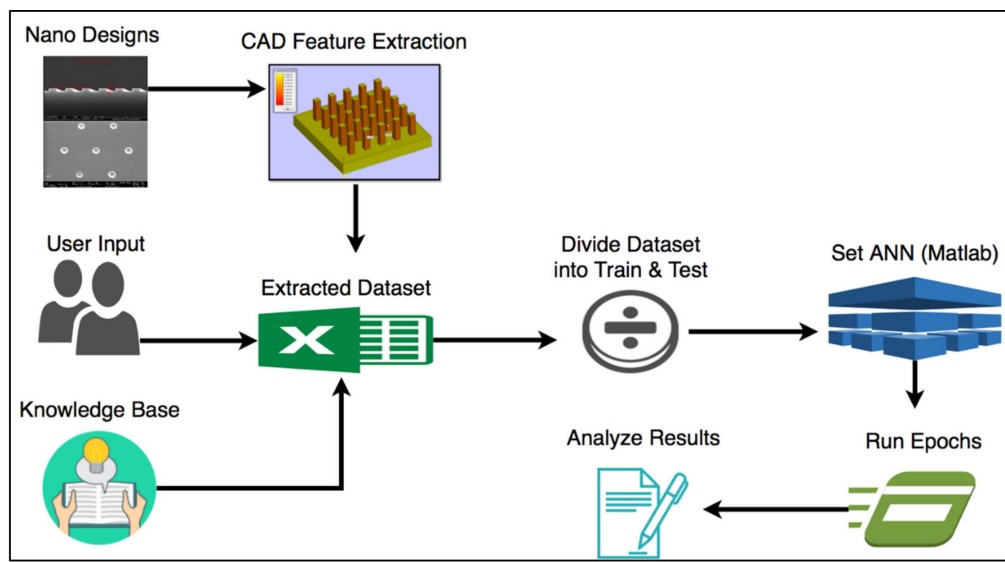

**Figure 3.** Flowchart to determine optimal NM processes (Authors' work).

### 2.1.1. Digital Designs and Feature Extraction

Digital designs of nano/micro-parts were retrieved using online resources, journal articles, and laboratory experiments [44–46]. Figure 4 shows the different types of nanoscale designs with varying levels of complexity. Materialize/Magics software package was used to extract geometric and topographical information from the CAD models. A total of 6 input variables were selected to evaluate the nanoscale designs for optimal process selection as shown in Tables 1 and 2. Three input variables, which include pattern complexity, aspect ratio, and feature resolution were evaluated using the CAD package. The material type and process throughput inputs were based on user preferences, whereas the fabrication and material costs were based on literature review and OEM specification sheets [47]. Figure 5 depicts the dimensional feature extraction procedure for a nanopillar array design in the Magics software package. These include volume, surface area, and specific dimensions of the features. Furthermore, a thickness analysis was conducted to determine the feature resolution.

### 2.1.2. Knowledge Base

A knowledge base was generated using literature reviews and best practices in the field [48]. In addition, process capability data were obtained from OEMs [49]. Table 1 shows the input variables with their respective keys. Table 2 shows the ranges of all the six input variables for five different nano/micro-manufacturing processes. These processes were chosen based on differences in their process capabilities such as feature resolution, throughput, processing cost, and applicability to manufacturing a variety of nano/micro-scale design components. Dip pen nanolithography (DPN) is a scanning-probe-based direct-write tool for generating surface-patterned chemical functionality on the sub-100 nm length scale [50]. It harnesses the power of an atomic force microscope (AFM) to deliver different ink formulations on substrates [51]. Nanoimprint lithography (NIL) is a high-production patterning tool for polymeric nanostructures with high precision and low cost. NIL achieves resolutions beyond the limitations set by light diffraction or beam

scattering that are encountered in conventional techniques [52]. Photolithography utilizes CAD-developed masks and ultraviolet light curing of polymers, which is a production tool for the semiconductor industry [53]. Pulsed laser deposition (PLD) is a bottom-up thin-film deposition technique that utilizes a plasma plume to deposit high-temperature superconductors, compound semiconductors, dielectrics, ferroelectrics, electro-optic and giant magnetoresistance oxides, polymers, and various types of heterostructures [54]. Self-assembly (SA) refers to the process by which nanoparticles or other discrete components spontaneously organize based on chemi-adsorption phenomena. Self-assembly is typically associated with thermodynamic equilibrium, and the organized structures are characterized by a minimum in the system's free energy [55].

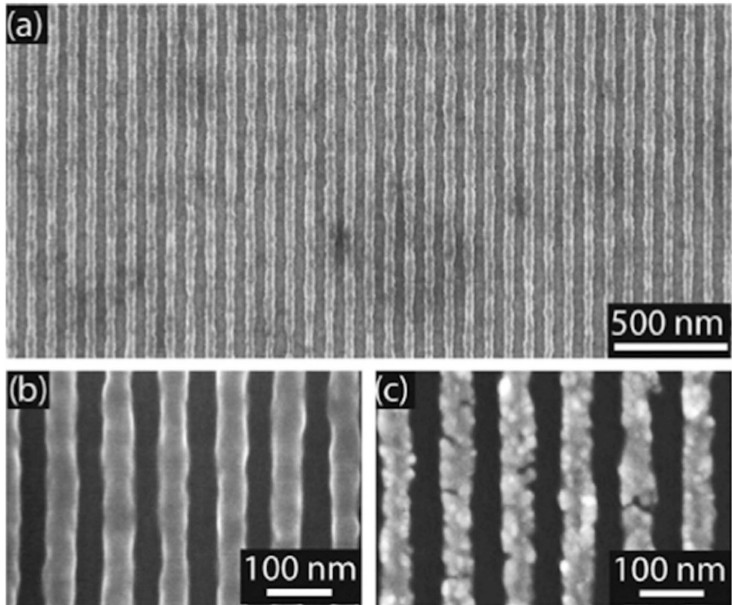

**Figure 4.** Nanoscale designs with varying levels of complexity: (**a**) Low magnification, (**b**) high magnification images of the pattern, and (**c**) silver lines after evaporation of 15 nm Ag and lift-off [44].

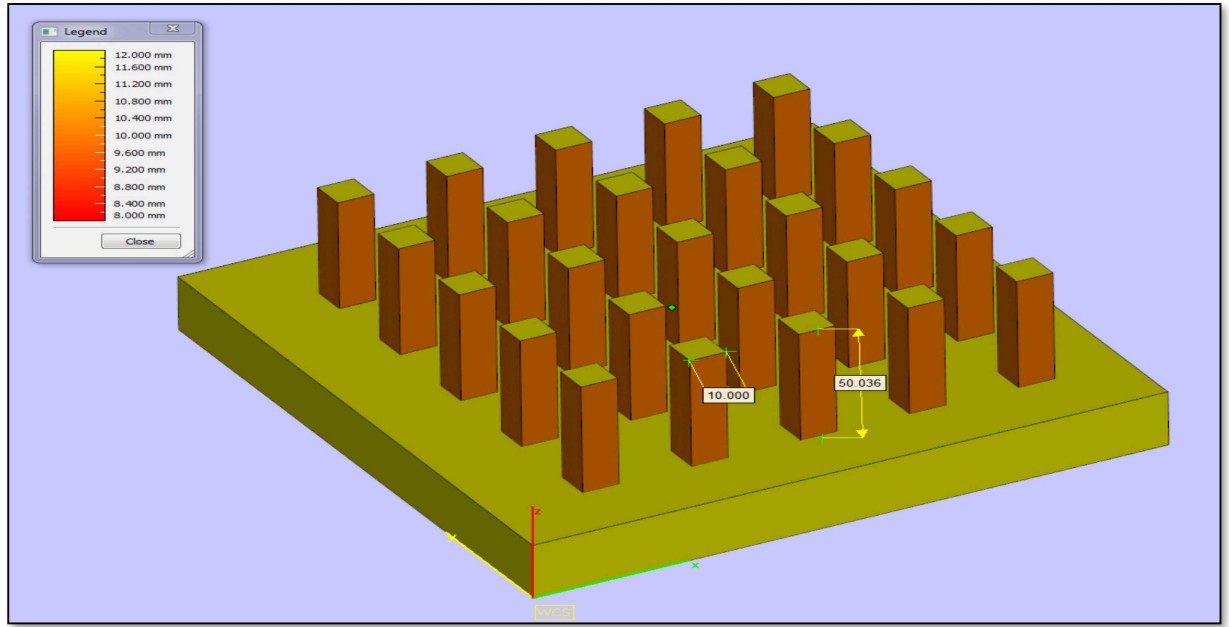

**Figure 5.** Dimensional and topological data extraction for datasheet (Authors' work).

**Table 1.** Keys for the input and output variables.

| Input/Output | Variables | |
|---|---|---|
| | **Cost** | |
| | Low | 1 |
| | Medium | 2 |
| | High | 3 |
| | **Pattern Complexity** | |
| | Low | 1 |
| | Medium | 2 |
| | High | 3 |
| Input | **Process Throughput Key** | |
| | Low | 1 |
| | Medium | 2 |
| | High | 3 |
| | **Material Key** | |
| | Polymer | 1 |
| | Metal | 2 |
| | Ceramic | 3 |
| | Semiconductor | 4 |
| | Composite | 5 |
| | **Nano-Process** | |
| | Dip Pen Nanolithography (DPN) | 1 |
| | Nanoimprint Lithography (NIL) | 2 |
| Output | Photolithography (PHO) | 3 |
| | Pulsed Laser Deposition (PLD) | 4 |
| | Self-Assembly (SA) | 5 |

**Table 2.** Features Ranges for the Nano-Processes.

| Input Variable | DPN | NIL | PHO | PLD | SA |
|---|---|---|---|---|---|
| Cost | [2, 3] | [1–3] | [1, 2] | [2, 3] | [1, 2] |
| Pattern Complexity | 2 | [2, 3] | [2, 3] | [1, 2] | [1, 2] |
| Aspect Ratio | [0.3–2] | [3–10] | [1–5] | [1–5] | [1, 2] |
| Feature Resolution (nm) | [5–20] | [5–20] | [500–800] | [5–20] | [10–50] |
| Process Throughput | [1, 2] | [2, 3] | 3 | [1, 2] | [1–3] |
| Material | [1–5] | [1–5] | [1–5] | [1–5] | [1–5] |

### 2.1.3. Data Organization

Based on inputs from the CAD feature extraction and user inputs, a data sheet was generated (Table 3). Expert opinion and inputs from the knowledge base were used to determine the appropriate nano/micro-manufacturing process for each part design.

**Table 3.** Data Sheet for the input dataset for ANN.

| Set | Cost | Pattern Complexity | Aspect Ratio | Feature Resolution | Process Throughput | Material | Nano-Process | Nano-Process Code |
|-----|------|--------------------|--------------|--------------------|--------------------|----------|--------------|-------------------|
| 1 | 1 | 2 | 8 | 12 | 2 | 4 | NIL | 2 |
| 2 | 1 | 2 | 8 | 6 | 3 | 1 | NIL | 2 |
| 3 | 1 | 1 | 2 | 34 | 1 | 4 | SA | 5 |
| 4 | 2 | 2 | 2 | 7 | 1 | 5 | PLD | 4 |
| 5 | 2 | 2 | 2 | 511 | 3 | 4 | PHO | 3 |
| 6 | 3 | 2 | 2 | 10 | 1 | 3 | PLD | 4 |
| 7 | 1 | 2 | 2 | 567 | 3 | 4 | PHO | 3 |
| 8 | 2 | 2 | 9 | 6 | 2 | 3 | NIL | 2 |
| 9 | 2 | 2 | 10 | 20 | 2 | 2 | NIL | 2 |
| 10 | 2 | 2 | 2 | 14 | 1 | 3 | PLD | 4 |
| 11 | 3 | 2 | 1 | 5 | 1 | 2 | DPN | 1 |
| 12 | 1 | 2 | 3 | 591 | 3 | 1 | PHO | 3 |
| 13 | 2 | 3 | 8 | 15 | 2 | 3 | NIL | 2 |
| 14 | 1 | 2 | 3 | 516 | 3 | 2 | PHO | 3 |
| 15 | 1 | 1 | 1 | 30 | 1 | 2 | SA | 5 |

2.1.4. ANN Algorithm Development

Due to the variety of nanomanufacturing techniques with similar properties, the decision to choose the appropriate one is complicated for manufacturers [56]. So, the classification of NM processes through the neural network under various criteria influences the decision by narrowing down the selection with high accuracy. In this research, the integration of an ANN with an expert system for nanomanufacturing classification was explored.

ANNs are very useful in a wide range of applications, including Data Mining, Text Mining, Signal Filtering, and Robust Pattern Detection, significantly improving their performances compared to the conventional techniques [57]. This system is versatile in operation, performs a broader range of functions compared to a linear program, and is robustly resistant to failure due to its parallel structure. This extremely reduces the processing time, unlike other kinds of algorithms, yet it achieves analogous results. Therefore, the application of ANN can be used successfully to support nanoscale manufacturing. It lowers costs and saves time. It can deal with issues that have a significant amount of data.

BPNN is one of the most common classes of training algorithms for Feed-Forward Neural Networks (FFNNs), also called Back Propagation (BP). It is one of the most popular NN methods and has four steps [58]. The BPNN corrects the randomly chosen weights. The four steps are the feed-forward computation, BP to the output layer, BP to the hidden layer, and weight updates. Once the error function becomes very small, the BPNN stops. The BPNN algorithm comes from a relatively simple idea: the output of the NN is evaluated against the expected output. The process is continuously repeated by modifying the connection (weights) between layers until the smallest possible error is obtained [59]. The output is computed by Equations (1)–(4):

$$O_i^d = f\left(net_i^d\right) = f(\sum_{j=1}^{n} W_{ij} V_j^d) = f(\sum_{j=1}^{n} (W_{ij}.f\left(\sum_{k=1}^{m} w_{jk} x_k^d\right))) \tag{1}$$

$$net_j^d = \sum_{k=1}^{m} w_{jk} x_k^d \tag{2}$$

$$V_j^d = f\left(net_j^d\right) = f\left(\sum_{k=1}^{m} w_{jk} x_k^d\right) \tag{3}$$

$$net_i^d = \sum_{j=1}^{n} W_{ij} V_j^d = \sum_{j=1}^{n} \left(W_{ij} \cdot f\left(\sum_{k=1}^{m} w_{jk} x_k^d\right)\right) \tag{4}$$

where $n$ is the number of inputs, $m$ is the number of hidden layer neurons, $d$ is the dimensional space, and $W_{jk}$ is the weight between neurons $j$ and $k$.

Probabilistic Neural Networks (PNNs) are the second algorithm that consist of 3 layers of neurons (nodes). The input layer consists of nodes (features). The hidden layer receives the features from the input layer. The third one is the output layer. PNN requires no time to train, and it is sensitive to outliers, so it can generate probability scores and approach Bayes optimal [48]. PNNs are generally considered classifier methods that represent any input pattern into several categories. They use a controlled training set to obtain the probability density functions improved in a pattern layer. The output layer is represented by Equation (5):

$$f(x) = \frac{1}{(2\pi)^{\frac{p}{2}} n\sigma^p} \sum_{i=1}^{n} \exp\left(-\frac{\|x - x_i\|^2}{2\sigma^2}\right) \tag{5}$$

where $n$ is the total number of training patterns, $d$ is the dimension of the input space, $\sigma$ is an arbitrary smoothing parameter (0 to 1), and $p$ is the dimensionality of measurement space.

Generalized Regression Neural Network (GRNN) is the third algorithm used, and it is a particular case of Radial Basis Networks (RBN). The GRNN architecture has two layers, which are the pattern and summation. Specht (1991) has claimed that the GRNN estimation is always able to converge to a global solution without being trapped by a local minimum [60]. GRNN is a neural network algorithm that uses input data to predict the output while requiring data for training. The predicted result or output is achieved by training the network with the data set, while also providing a new testing data set. The output is calculated as a function of the weighted average of the training data set [61], where ($Y(x)$) is the prediction value of input $x$ as shown in Equations (6) and (7):

$$Y(x) = \sum_{k=1}^{n} \left(y_k * K * exp^{\frac{-d_k}{2\sigma^2}}\right) / \left(\sum_{k=1}^{n} K * e^{\frac{-d_k}{2\sigma^2}}\right) \tag{6}$$

$$d_k = (x - x_k)^T (x - x_k) \tag{7}$$

where $y_k$ is the activation weight for the pattern layer neuron at $k$, $d_k$ is the squared Euclidean distance between the training samples $x_k$ and the input $x$, and $T$ is a threshold or bias value.

MATLAB is supposed to be the best computing engine concerning its ease of use and fast speed for GRNN implementation [62]. MATLAB develops GRNN and identifies an optimal value of Sigma using split-sample cross-validation [63]. As a result, MATLAB was used for classification, and the Levenberg–Marquardt training algorithm was applied for training the feed-forward network. The validation of the output was compared with the target based on desired NM processes. MATLAB codes helped in developing ANN models for the prediction of nano-processes. Scripts can be developed to build commands and functions to customize training algorithms and test the performance of the ANN models [18]. Therefore, all the inputs and outputs were fed into the datasheet, and MATLAB software was used to build neural network codes for three different algorithms (GRNN, BPNN, and PNN).

### 2.2. Cyber Interface Simulator Using the IoT

The IoT has been used in this research to simulate machine availability for each NM process based on the ANN outputs. Node-Red is a web-based tool that connects hardware devices and an Application Program Interface (API). An API represents the interaction of software components. It provides a browser-based flow editor as well. It is a powerful

tool for building IoT applications with a focus on simplifying the 'wiring together' of code blocks that carry out the tasks [24]. Node-Red adopts a visual programming method that allows developers to connect already defined code blocks, known as 'nodes,' together to perform a task. When wired together, input nodes, processing nodes, and output nodes combine to make up 'flows', which are deployed to the runtime in a single click. With JavaScript, functions can be created within the editor using a rich text editor; a built-in library also makes it possible to save useful functions, templates, or flows for reuse [64]. Therefore, the Node-Red IoT simulator was chosen to create a Cyber Interface Simulator as in Figure 6. The input was provided to initiate time for 2 h to check the availability of nanomachines on the cyber network. A function generator was coded using JavaScript. A dashboard interface was established which received input from the function generator to present useful data using different graphical tools. It momentarily provides the actual number of available machines, including their kinds, in cyberspace. Machine availability arrays were automatically created and saved for all simulation periods.

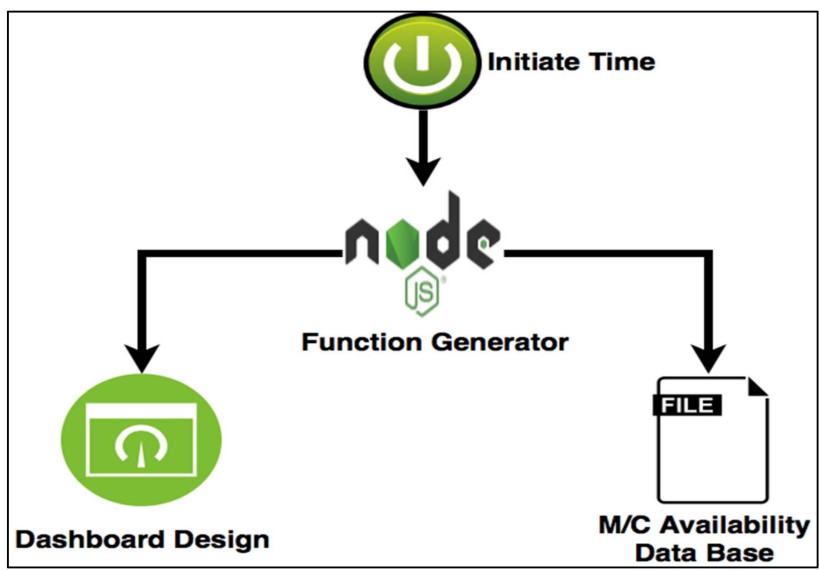

**Figure 6.** Framework for Cyber Interface Simulator using Node-Red.

### 2.3. Dynamic Nano-M/C Identification System

The ANN provided the predicted NM processes, whereas the Cyber Interface Simulation (Figure 7) provided the availability of nanomachines. However, to check the compatibility of the two outputs, they need to be handled and controlled, requiring time and expenses to check the two expert systems and identify the final assignment. Therefore, a dynamic system is appropriate and suited to perform this task. It finds out what the right allocation of the machine availability is and the accuracy of the assignment. A dynamic system is an integration method providing the percentage of assignment and the machine allocation details. In the received the predicted nano processes from the ANN and the machine availability from the Node-Red. Then, it compared them to compute the final machine availability assignment in addition to the ANN's accuracies. Therefore, it provided a chart of their accuracy percentages.

The Java programming language was used to code this system. The input part consists of the ANN and Node-Red machine availability arrays. The system analyzed the input and produced the percentage accuracy of the assignment. Since the PNN algorithm has the highest average prediction accuracy of 96%, it was run 14 more times. Therefore, the PNN had 24 accuracies including the correct classification processes. The machine type availability of 24 periods was provided by the Node-Red in array format. A Java-based code was developed to compare every period separately. Each period had the correct classification processes from the ANN and the type of available machines. It compared every process with its available machines and chose the one with fewer. Similarly, it did

it for the rest of the processes and their machines to obtain a final column of one period. Then, it shows the percentages of the final assignment and the ANN output.

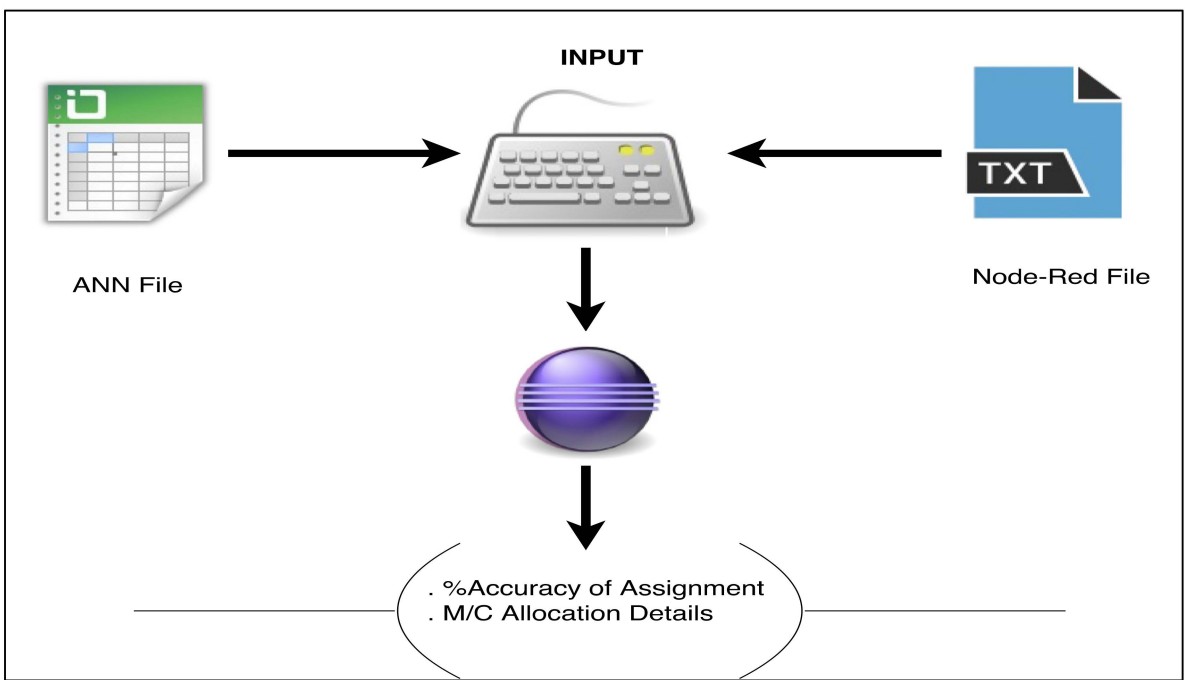

**Figure 7.** Dynamic Nano-M/C Identification System (Authors' work).

### 3. Result and Discussion

The input consisted of different designs (nano/micro) and user inputs. ANN algorithms were implemented to select the optimal nano/micro-manufacturing process. A cyber interface was modeled using the Node-Red IoT simulator, which identified machine capability and availability on the network. A dynamic machine identification code was developed using the Java programming language to predict real-time machine assignment.

### 3.1. ANN-Based Expert System

#### 3.1.1. First Stage

One hundred designs were used with only four inputs (features). These inputs were pattern complexity, feature resolution, process throughput, and material type. Thus, it is considered to be the first stage of improvement (Table 4). These input variables were chosen as they represented important aspects while selecting a suitable nano/micro-manufacturing process for the given design input. The neural network models were configured in MATLAB software, that provides a platform for the simulation application. The dataset was divided into input vectors and target vectors as follows: 75% were used for training, and 25% were used as an entirely independent test of network generalization. The Levenberg–Marquardt (LM) algorithm is an algorithm that trains a neural network. The network training function updates weight and bias values according to the LM algorithm. Although LM requires more memory than other algorithms, it is the fastest supervised algorithm [65]. LM is a variation of Newton's method that was designed for minimizing functions that are sums of squares of other nonlinear functions. This is very well suited to neural network training, where the performance index is the mean squared error. Three neural network algorithms were implemented, which include GRNN, PNN, and BPNN. The data were set up for a neural network by organizing the data into two matrices: the input matrix and the target one. The input matrix had elements in columns representing selected features for the collected nano-designs in rows. The output (target) part was a column that included the number of designs for five different processes (20% of all designs are related to one process).

**Table 4.** ANN stages.

| Stage | Designs | Input Variables |
|---|---|---|
| First | 100 | Pattern Complexity |
| | | Feature Resolution |
| | | Process Throughput |
| | | Material |
| Second | 200 | Pattern Complexity |
| | | Feature Resolution |
| | | Process Throughput |
| | | Material |
| | | Cost |
| | | Aspect Ratio |

The network architecture has an input layer with several features, one hidden layer with four nodes, and five output layers. The numbers of nodes in the hidden and output layers were chosen after performing a sensitivity analysis for different combinations of hidden and output layer networks. The number of input neurons was set to the number of the design features, whereas the hidden neurons were set to four. The number of output neurons was set to five, which is equal to the number of (processes) targets. The neurons' connection strength is based on their weights.

Figure 8 shows the best run in the first stage, which is related to the PNN since the overall progress of ANN is different where the training stopped on reaching the maximum validation check of 6 at iteration 10. The performance (MSE) was 0.0560, the gradient descent value was 0.0069, and the mu value was $1.00 \times 10^{-7}$. The training continued through 10 epochs but the best validation performance was reached at epoch 4 with a value of 0.082469. The regression plot of training, test, validation, and the training R-value was 0.73684, the validation R-value 0.69892, the test R-value 0.73742, and the overall R-value equal to 0.73107. This proves the developed model and the network procedure of training, testing, and validation are acceptable but not satisfactory.

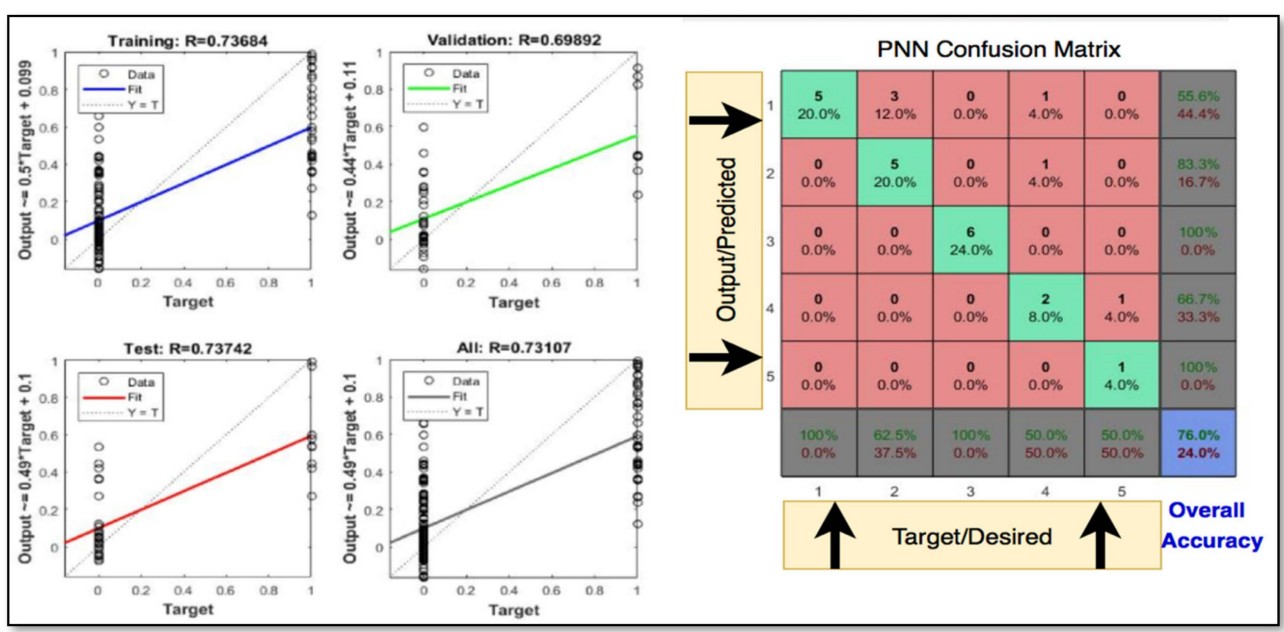

**Figure 8.** The First Stage best run PNN (Authors' work).

The output class was our PNN prediction values, and the target class was our original output values, as in the confusion matrix (Figure 8). Out of nine DPN predictions, 55.6% were correct and 44.4% were wrong. Out of six NIL predictions, 83.3% were correct and 16.7% were wrong. Six PHO predictions were 100% correct. Out of three PLD predictions, 66.7% were correct and 33.3% were wrong. One SA prediction was 100% correct. Overall, 76% of the predictions are correct and 24% are wrong classifications, but the average accuracy of the 10 runs is 64%.

The ANN computed the prediction accuracy by dividing the total corrected outputs by the total of the test input (target). Table 5 shows the prediction accuracies of the 10 runs for the first stage. PNN has the highest average accuracy of the first stage, which is 65.20%, whereas the GRNN and BPNN have 64% and 62.80%, respectively. However, the maximum PNN accuracy is 76%, whereas the minimum is 48%.

**Table 5.** First Stage Result.

| | First Stage | | |
|---|---|---|---|
| | **GRNN** | **PNN** | **BPNN** |
| 1 | 64% | 48% | 60% |
| 2 | 72% | 64% | 68% |
| 3 | 60% | 64% | 56% |
| 4 | 64% | 72% | 56% |
| 5 | 72% | 64% | 56% |
| 6 | 64% | 64% | 64% |
| 7 | 68% | 76% | 64% |
| 8 | 68% | 64% | 68% |
| 9 | 44% | 76% | 68% |
| 10 | 64% | 60% | 68% |
| Average | 64% | 65.20% | 62.80% |

### 3.1.2. Second Stage

In the second stage, the accuracy of the ANN was improved by adding two more input variables, which included the cost and aspect ratio. In addition, the dataset was doubled to input 200 designs. Thus, there were a total of six input features to evaluate the 200 designs (Table 4), giving the network a higher discriminating ability to distinguish between the input requirements and map them to accurate output processes. The enhanced ANN model gave significantly higher prediction accuracies for each of the algorithms (Figure 9).

After running the ANN 10 times, 10 results were shown for GRNN, PNN, and BNN. The PNN prediction accuracy average is the highest at 96%. The three averages are better than the ones in the first stage. However, the results of the best runs for PNN had a high correlation coefficient (R-value = 0. 9817) between the measured and predicted output variables, as shown in Figure 9. In addition, the PNN confusion matrix in Figure 10 shows that the overall accuracy is 98%, where there is only one misclassification, in which the ANN wrongly assigned a NIL process as a PLD.

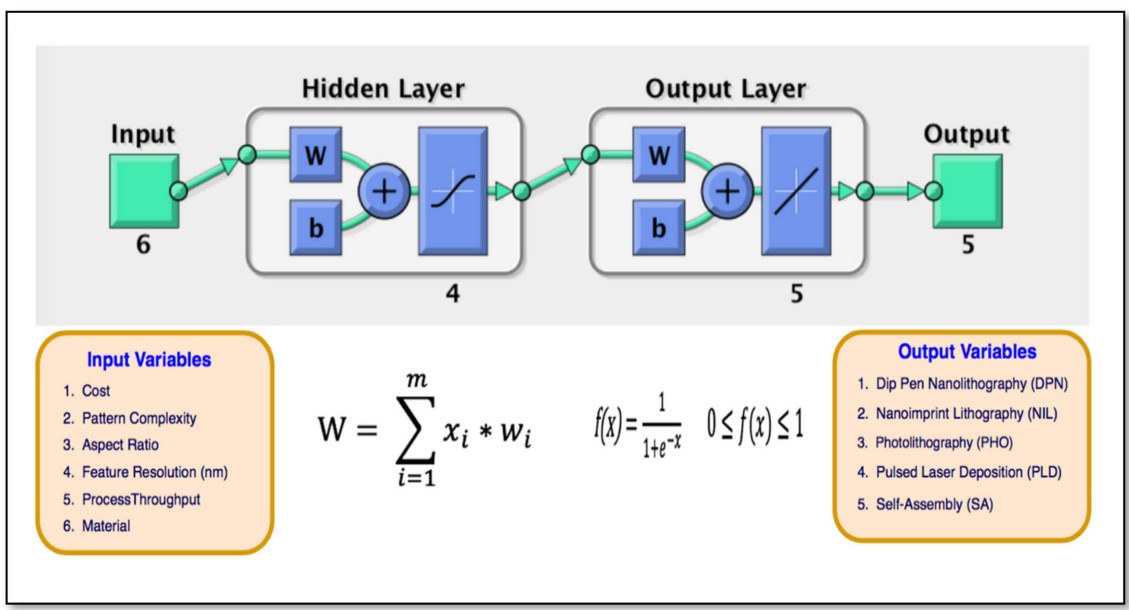

**Figure 9.** Second-stage ANN model with input variables, hidden layers, and output configuration (Authors' work).

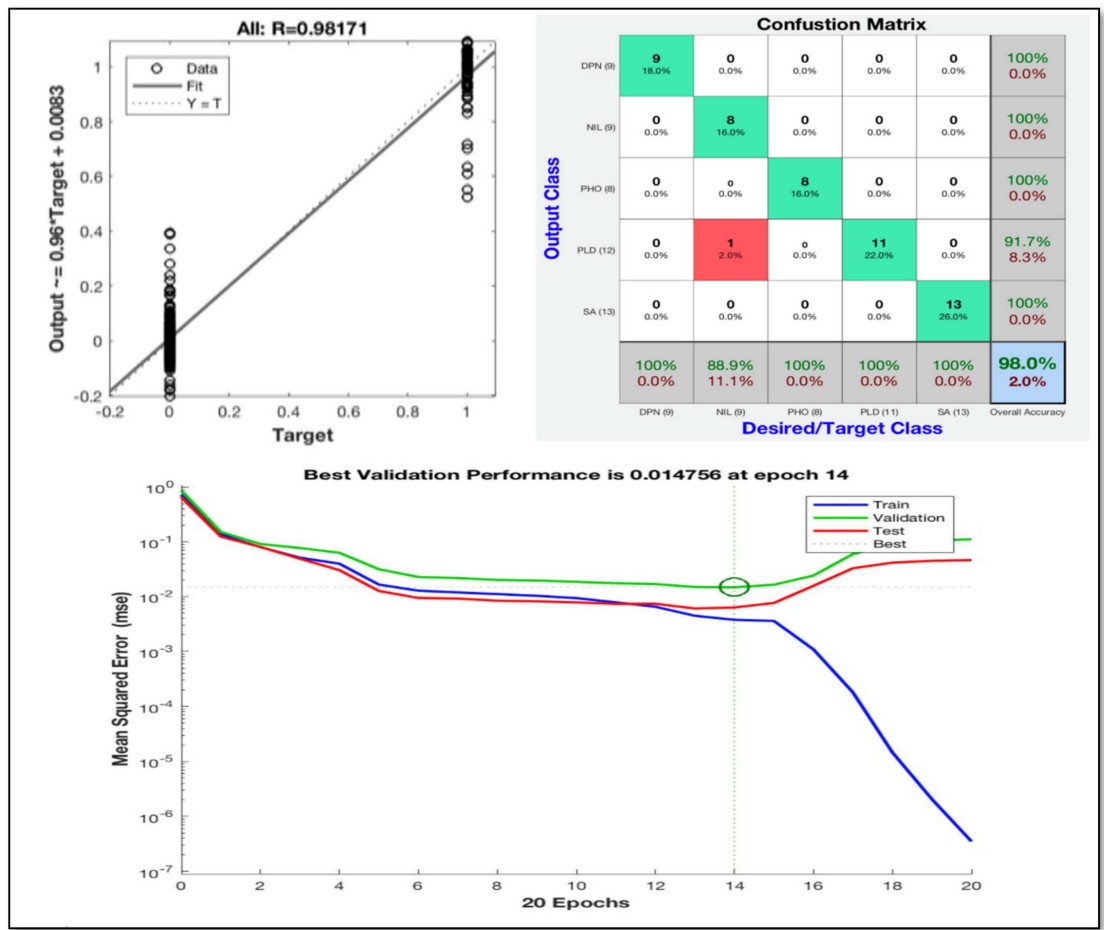

**Figure 10.** The second stage's PNN best run result (Authors' work).

Table 6 shows the 10 run results of the second stage. The testing part shows that the highest average accuracy is the PNN one (96%), while the GRNN and BPNN have accuracies of 95% and 94.20%, respectively.

**Table 6.** Second Stage Result.

| | Second Stage | | | | | | |
|---|---|---|---|---|---|---|---|
| | Training | | | | Testing | | |
| Run | MSE | Gradient | Validation Performance | R-Value | GRNN | PNN | BPNN |
| 1 | $4.12 \times 10^{-7}$ | 0.00015 | 0.01455 | 0.9800 | 98% | 96% | 96% |
| 2 | $3.84 \times 10^{-7}$ | 0.00012 | 0.01421 | 0.9820 | 98% | 96% | 98% |
| 3 | $3.51 \times 10^{-7}$ | 0.00014 | 0.01475 | 0.9817 | 96% | 98% | 96% |
| 4 | $3.63 \times 10^{-7}$ | 0.00012 | 0.01460 | 0.9811 | 96% | 98% | 96% |
| 5 | $3.86 \times 10^{-7}$ | 0.00022 | 0.01399 | 0.9700 | 96% | 98% | 90% |
| 6 | $3.75 \times 10^{-7}$ | 0.00018 | 0.01400 | 0.9788 | 94% | 96% | 96% |
| 7 | $3.72 \times 10^{-7}$ | 0.00020 | 0.01414 | 0.9786 | 96% | 94% | 96% |
| 8 | $3.88 \times 10^{-7}$ | 0.00019 | 0.01512 | 0.9814 | 94% | 96% | 94% |
| 9 | $3.98 \times 10^{-7}$ | 0.000276 | 0.01537 | 0.9686 | 90% | 94% | 90% |
| 10 | $3.87 \times 10^{-7}$ | 0.000250 | 0.01477 | 0.9733 | 92% | 94% | 90% |
| | Average | | | | 95.00% | 96.00% | 94.20% |

### 3.2. Cyber Interface Simulator Using IoT

After implementing the ANN expert system for evaluating appropriate nano/micro-manufacturing processes for each part's design, it is important to next seek the availability of machines on the cyber network. The Node-Red IoT simulator was implemented to accomplish this task. As per the framework for Cyber Interface Simulator using Node-Red discussed in Figure 5, the Node-Red program was developed to execute the machine availability function [66].

Different nodes, as shown in Figure 11, were used to simulate the cyber interface flow. First, the inject node was added as an input by dragging it onto the workspace from the palette. The input node initiates the flow for a given interval of time depending on the application intent. The timestamp was triggered every 3 s to simulate each period of machine availability. In practice, this would relate to pinging different machines that are configured on a central hub for an interval of time (e.g., every 2 or 3 h).

Second, the function node was used and fed by the correct code to find the available number of machines and their classifications. The function node activated messages through a JavaScript function. It is called a machine availability generator and was coded to generate a random number between 30 and 50. The output debugs node, the total number of M/C available, shows the machine's available number during a specific time on the debug tab. In addition, the M/C availability status node, output debug, was used to provide an array of the available machines related to the total number of M/C available nodes. However, dashboard nodes were used to provide the needed output. Thus, the gauge node, which controls the number of machines available, was modified so that the range was from 30 to 50. Therefore, a file of any period can be usefully used since it has arrays of machines' availability. So, a file node, machine file, has been added to provide a file of 24 times (time ranges? time intervals?) of the available machines, and this can be modified. This was formed as 24 arrays, where each array included the machine's number every 3 s.

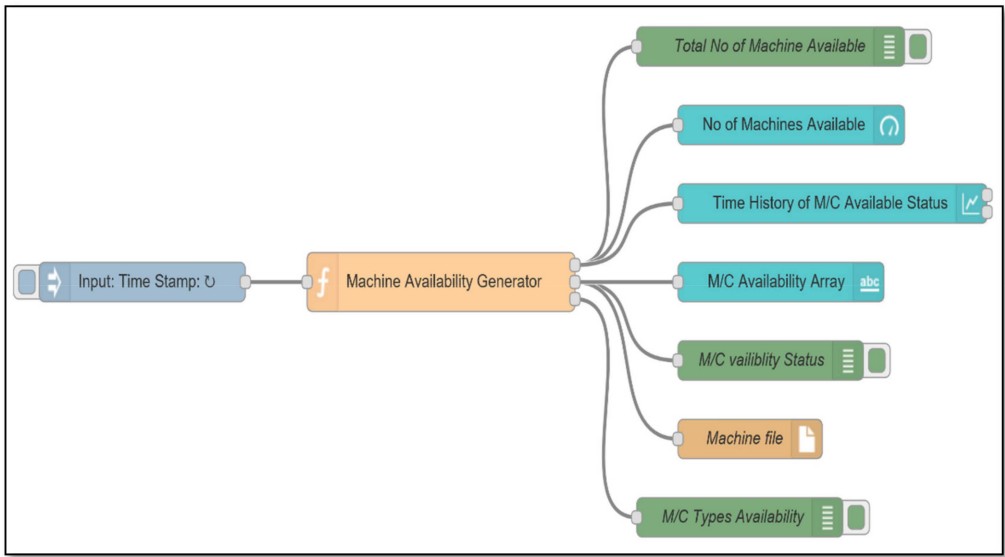

**Figure 11.** Node-Red cyber interface flow (Authors' work).

The machine availability array, text node, has been used to classify the machines. Then, the nodes were wired together by dragging between them. The debug console outputted the total number of machines available and the machine available status. The number of machines available (gauge) provides the last available number on the dashboard. In addition, the chart shows the time history of machine available status where the *y*-axis represents the number of machines, and the *x*-axis is the time. Finally, the dashboard shows up with the gauge and the chart (Figure 12). The machine file nodes create the final file of machine type's arrays that the machine-type array node performs every period of time.

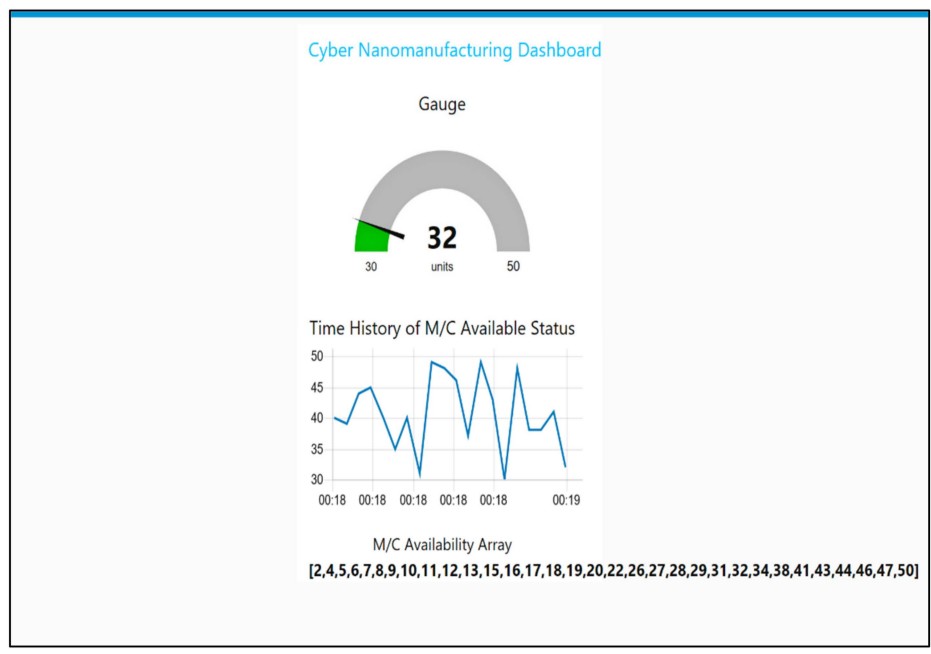

**Figure 12.** Cyber nanomanufacturing dashboard interface (Authors' work).

### 3.3. Dynamic Nano-M/C Identification System

The Dynamic Nano-M/C Identification System was used to integrate the ANN-Based Expert System and the Cyber Interface Simulation to compute the final accuracy of the assignment. The ANN results are the predicted nano-processes, whereas the Node-Red result is the machines' availability. The Java programming language was the tool used

in this system. The proper code has been developed to perform the needed assignment. First, it classified the Node-Red arrays one by one based on Figure 13 to provide a new file having arrays of five different machines. Then, it compared the new file with the ANN file row by row. It compared the number of processes and their machines and chose the minimum one so that the final assignment could be computed.

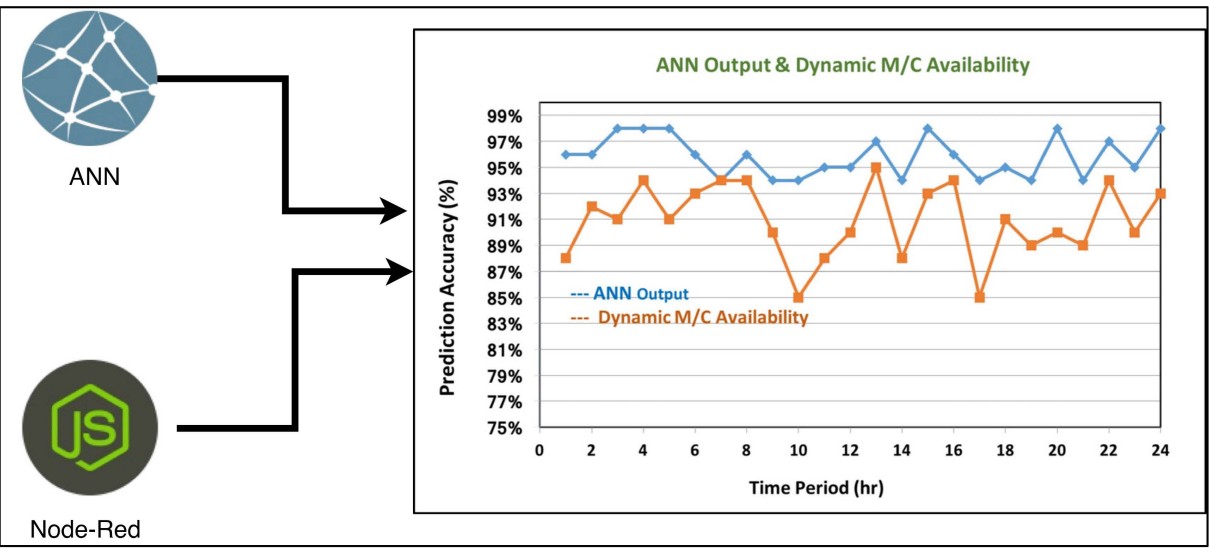

**Figure 13.** Steady-state ANN vs. Dynamic (Authors' work).

Next, it calculated the total for every row and divided it by the maximum number of machines available, i.e., 50. Therefore, 24 columns of percentages were computed to represent the final assignment (dynamic). On the other hand, the ANN file has been used to maintain a steady state. This was performed by computing the total for every row and dividing it by 50. Since the new two columns of percentages are ready, the chart has been created to show the dynamic and the steady-state flow as in Figure 13.

## 4. Conclusions

The revolution in digital technology, IoT, and nano/micro-manufacturing technologies offer added advantages if they are interfaced in real-time over a cyber network. Many industries operating on nano/micro-manufacturing technologies work in standalone configurations, which prevents the system integration benefits. The present research focuses on developing a smart agent system with predictive capabilities for cyber nanomanufacturing in real-time. The developed framework helps in translating nano/micro-scale digital design to appropriate process selection and machine availability using smart algorithms. Various AI-based technologies (GRNN, BPNN, and PNN), along with IoT-based technologies (API, Cyber Interface Simulator, and Node-Red), have been integrated. The developed smart agent system was successfully tested on various nano/micro-manufacturing technologies such as DPN, NIL, PHO, PLD, and SA. Thus, the developed dynamic nano-M/C identification system helps in identifying machine availability over the cyber network in real-time.

In the present research setup, a maximum of 50 machines are made available. The ANN output and dynamic machine availability were determined. A maximum prediction accuracy of 99% was obtained for the ANN base output, while a 95% prediction efficiency was obtained for dynamic machine availability. Future research may explore the parametric optimization of various nano/micro-manufacturing equipment while selecting machine availability to optimize the production efficiency and reduce machine idleness.

**Author Contributions:** Conceptualization, N.A. and S.D.; Investigation, S.D., S.A. and M.R.N.M.Q.; Methodology, N.A., S.D. and S.A.; Resources, S.A. and M.R.N.M.Q.; Software, N.A. and S.D.; Supervision: S.A. and M.R.N.M.Q.; Writing—review & editing, N.A. and M.R.N.M.Q. All authors have read and agreed to the published version of the manuscript.

**Funding:** This research was funded by the Deanship of Scientific Research, King Khalid University, Kingdom of Saudi Arabia, and the grant number is R.G.P. 1/212/41.

**Institutional Review Board Statement:** Not applicable.

**Informed Consent Statement:** Not applicable.

**Data Availability Statement:** Not applicable.

**Acknowledgments:** We would like to express our gratitude to the Deanship of Scientific Research, King Khalid University, Kingdom of Saudi Arabia for funding this work, as well as family, friends, and colleagues for their constant inspiration and encouragement.

**Conflicts of Interest:** The authors declare no conflict of interest.

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
