# Peer review of "Smart Agent System for Cyber Nano-Manufacturing in Industry 4.0"

_applsci, doi:10.3390/app12126143_

Round 1
Reviewer 1 Report
The manuscript "Smart Agent System for Cyber Nano Manufacturing in Industry 4.0" describes This paper describes the development of a cyber nanomanufacturing framework to integrate the design
and manufacturing of nanoscale components/devices over cyberspace. The framework consists of three sub systems which include: (1) Artificial Neural Network (ANN) Based Expert System, (2) Cyber Interface Simulator (IoT), and (3) Dynamic Nano M/C Identification System.
This is a relevant review and can be accepted after the following comments.
The abstract is too wide and can be trimmed.
Authors used SEM images in Fig 3. This image needs proper visible scale bars. it is not clear the dimensions of the features.
If Fig. 3 is not the author's original work, needs to cite actual literature from where it is used.
The introduction section is lacking a future vision of why this research is important and what would be its future implications.
Conclusion is not reflecting proper impact and novelty of the work. It is too broad. Author may reconsider rewriting the conclusion sections as well.
Author Response
We upload the corrected version of “Smart Agent System for Cyber Nano Manufacturing in Industry 4.0” .
We thank you very much for your valuable time and comments, which have enhanced the quality of the present manuscript considerably.

Reviewer 2 Report
Ok for publication.
Author Response

(The authors gave the same response as above.)

Reviewer 3 Report
Please check text formatting lines 66-70
Check the unifinished sentence lines 72-73
Please check the spelling and grammar
Check lines 182 and 184
Fix Table 4 design
Can you compare your results with similar models previously developed?
Author Response

(The authors gave the same response as above.)

Reviewer 4 Report
This manuscript is potentially of interest to readers of the journal, however some parts need to be revised before publication.
-
The topic of smart manufacturing in the context of Industry 4.0 (and its evolution) requires an overview consistent with the latest scientific research findings. It is therefore imperative that the authors revise the introduction by updating the literature review considering significant recent studies on the topic. The bibliography proposed by the authors is built with very old articles and none after 2020, which is unusual for a topic (considered in this research) that evolves day by day.
- In line 116 the authors identified a GAP that their research should fill: "There is currently no system for accurately identifying and efficiently matching available resources with needs." This statement must refer to recent and authoritative literature to make the theoretical basis of the manuscript sufficiently sound and to justify the purpose of the study.
- Section 2 (Methodology) does not describe the real methodological approach of the research, but presents a set of techniques while also anticipating some results. The authors should present to the readers the methodology they have adopted justifying their choice with references to recent literature. In addition, the results should all be included in a specific section.
- At the conclusion of the methodology section and before the results, authors should convince readers, with solid arguments, that their research is innovative compared to the state of the art because it fills a knowledge gap. It would also be appropriate to formulate a few research questions that should then be discussed in the conclusion.
- The conclusions do not include the scientific contribution made by this study to knowledge on the topic, as emerged from the state of the art in the literature review, as well as the operational implications for practitioners. Also missing are the limitations of this study and future lines of research.
Author Response

(The authors gave the same response as above.)

Round 2
Reviewer 4 Report
In this latest version of the manuscript, the authors did not implement the improvements suggested to them by the reviewers in particular:
(1) The inclusion of four new references are not sufficient to provide an overview of the state of the art of Industry 4.0 and technologies. Therefore, the introduction is still incomplete and especially not up-to-date with recent literature.
(2) In line 84, the authors did not link GAP to the literature as had been suggested to them, so this statement remains their opinion not scientifically validated.
(3) The Methodology section has not been modified as suggested, moreover, the authors still have not provided solid justifications for considering their study innovative compared to the state of the art.
(4) Lacking an actual theoretical GAP and a baseline on the state of the art, the conclusions even in the latest version, remain generic.
(5) Finally, I advise the authors to follow more scrupulously the directions of the reviewers which are always aimed at improving the quality of the manuscript to accompany its publication and to argue in more detail the response to the reviewers unlike what was done in this case instead.
Unfortunately, I have to conclude that this version of the manuscript is not suitable for publication since the major I had previously assigned was a gesture of goodwill in the hope of greater commitment on the part of the authors to improve their work.
Author Response

(The authors gave the same response as above.)
